# Association Between Working Memory at Age 4 Years and Night Sleep Duration and Yogurt Intake Frequency at Age 1 Year

**DOI:** 10.3390/nu17193081

**Published:** 2025-09-27

**Authors:** Yuki Otsuka, Shoji Itakura, Motonobu Watanabe, Kumiko Kanatani, Kyoko Hirabayashi, Fusako Niwa, Takeo Nakayama

**Affiliations:** 1Faculty of Psychology, Otemon Gakuin University, Ibaraki 567-8502, Japan; 2Research Organization of Open Innovation and Collaboration, Ritsumeikan University, Osaka 567-8570, Japan; 3Center for Baby Science, Doshisha University, Kizugawa 619-0225, Japan; 4Department of Pediatrics, NHO Minami Kyoto Hospital, Joyo 610-0113, Japan; 5Graduate School of Medicine, Kyoto University, Kyoto 606-8501, Japan; 6Department of Pediatrics, Mitsubishi Kyoto Hospital, Kyoto 615-8087, Japan

**Keywords:** working memory, executive functions, sleep duration, yogurt intake frequency, child development

## Abstract

**Background/Objectives**: This cohort study examined the effects of sleep durations (night, day, and total daily) at ages 1, 1.5, and 3 years on working memory (WM) assessed at age 4, measured using forward digit span. **Methods**: Because frequency of yogurt intake at 1 year has been shown to affect sleep duration at 3 years, we also accounted for the association between frequency of yogurt intake at 1 year and sleep duration, based on a recent study indicating positive effects of yogurt on sleep. The study included 164 mother–child pairs observed from ages 1 to 4. **Results**: Spearman correlations and hierarchical multiple regression analyses revealed that both night sleep duration and yogurt intake at age 1 were significantly associated with WM performance at age 4. In this sample, however, yogurt intake was not correlated with sleep duration. **Conclusions**: To a limited degree, both yogurt intake frequency and night sleep duration at 1 year were associated with WM performance at 4 years, indicating that frequent yogurt intake at 1 year and longer night sleep duration may lead to higher WM performance at 4 years.

## 1. Introduction

Among the cognitive functions that develop during childhood, executive functions are particularly important [1,2]. These functions are considered a prerequisite for higher intellectual activity and serve as determinants of the extent to which adults can eventually perform complex cognitive tasks [3,4]. In examining preschool-aged children’s development of executive functions, considering working memory (WM), the earliest developing component for executive function, is especially important [5]. WM begins to develop in infancy and improves throughout preschool age, enabling children to retain more information (for a review, see [6]).

Sleep is an essential activity for humans; it influences the development of brain functions and structures [7,8]. As a major biological factor, sleep also affects the development of executive functions, along with other biological and environmental factors associated with executive function performance [9]. Indeed, several studies have indicated that sleep disturbances are related to the impairment of executive function [10,11], and self-reported habitual sleep disturbances are linked to executive functions in early primary school children [12]. Furthermore, some evidence suggests that napping facilitates explicit memory consolidation (for a review, see [7]).

Prior studies have used the subjective measure of parent-reported sleep duration to examine WM at preschool age, finding that a longer night sleep duration may lead to improved WM performance [13,14]. However, other studies have reported that neither total nor night sleep duration significantly affects WM performance and that children who do not nap may perform better on WM tasks [15]. Moreover, some evidence has suggested that WM performance is lower in children with a shorter night sleep duration but a long day sleep duration [16]. Although napping has been reported to facilitate explicit memory consolidation, previous WM studies have reported negative effects on children’s WM performance [7].

While sleep is associated with executive function performance [9], recent studies have also indicated the importance of considering the impact of other factors. For example, Inoue et al. discovered that at age 1 year, the frequency of intake of yogurt, a fermented food, affects sleep duration at age 3 [17]. This relationship may be explained by the gut–brain axis, a bidirectional communication system between the gastrointestinal tract and the central nervous system. Yogurt, as a source of probiotics, can positively influence gut microbiota composition, which in turn potentially affects sleep patterns and cognitive functions, including WM [18,19,20]. Despite increasing academic interest in the gut–brain axis and its potential influence on cognitive development, no previous study has investigated how yogurt intake—a dietary factor known to affect gut microbiota—interacts with sleep duration to influence children’s WM performance. This study aims to address this critical gap in the literature.

Based on previous findings, we hypothesize that yogurt intake at age 1 year positively influences sleep duration at later ages and that a longer sleep duration is associated with better WM performance at age 4 years. The study aimed to examine these relationships using longitudinal data and behavioral assessments. More specifically, the study examined the effects of day, night, and total sleep duration at age 1, 1.5, and 3 years, respectively, on WM at age 4. The analysis also considered the effect of yogurt intake frequency at age 1 year on measures of sleep duration. We used digit span as a WM measure at 4 years and the communication subscale in the Ages and Stages Questionnaires (ASQ-3) as a measure of verbal ability at ages 3 and 4 years because it is highly likely that language development also impacts digit span performance. Following Bernier et al. [13], we conducted a hierarchical multiple regression analysis, controlling for the influence of verbal ability to examine the association between WM and sleep duration at preschool age. The aim of this study was to investigate the longitudinal associations between sleep duration at ages 1, 1.5, and 3 years and working memory performance at age 4, as well as the association between yogurt intake frequency and sleep duration.

## 2. Materials and Methods

### 2.1. Participants

The sample included 165 mother–child dyads (81 boys, 84 girls) who participated in the Sub-Cohort Study of the Kyoto Regional Centre of the Japan Environment and Children’s Study (JECS) [21]. The Sub-Cohort Study initially included 169 participants at age 4 years; however, 4 children who did not complete the digit span task of the Kyoto Scale of Psychological Development 2001 were excluded from the present study. All the children’s scores on the digit span task of the Kyoto Scale of Psychological Development 2001 at age 4 years were registered. This Adjunct Study analyzed digit span task scores extracted from the raw data of the Sub-Cohort Study of the Kyoto Regional Centre from a new perspective. We excluded one dyad with missing responses to the sleep questionnaire at age 1.5 or 3 years, ultimately analyzing 164 samples.

Parents’ educational levels were reported in the JECS, and annual household income was obtained in the questionnaire administered at age 3 years (all values in Japanese yen): less than 2 million, between 2 and 4 million, between 4 and 6 million, between 6 and 8 million, between 8 and 10 million, between 10 and 12 million, between 12 and 15 million, between 15 and 20 million, and more than 20 million.

### 2.2. Procedure

The JECS protocol was reviewed and approved by the Ministry of the Environment’s Institutional Review Board on Epidemiological Studies and all participating institutions’ ethics committees. This Adjunct Study was approved by the Ethics Committee of Doshisha University (23009) and Kyoto University (R4435). The participating mothers provided written informed consent. The JECS was conducted in accordance with the Declaration of Helsinki and all other national regulations.

The mother–child dyads participated in this study when the children were ages 1 (T1; M = 11.8 months, SD = 1.4), 1.5 (T2; M = 17.7 months, SD = 1.1), 3 (T3; M = 35.8 months, SD = 1.8), and 4 years (T4; M = 48.2 months, SD = 0.6). At T1, the mothers completed questionnaires on sleep and yogurt intake. At T2, they completed the sleep questionnaire. At T3, they completed the sleep questionnaire and the Ages and Stages Questionnaire-3 (ASQ-3). At T4, the children performed the digit span task of the Kyoto Scale of Psychological Development 2001 at the Kyoto Regional Centre. All measures were derived from the JECS and have been widely used in similar large-scale cohort studies [21].

To ensure adequate statistical power, we conducted a priori power analysis using G*Power software (version 3.1). The analysis indicated that a sample size of 164 participants is sufficient to detect a medium effect size (ρ = 0.3) in correlation analyses with 80% power at an alpha level of 0.05. This confirms that our study is appropriately powered to detect meaningful associations among the variables of interest. For all analyses, we used SPSS version 29 (IBM Corp., Armonk, NY, USA). Prior to handling missing data, we assessed all variables’ normality using both the Shapiro–Wilk test and visual inspection via Q–Q plots. The Shapiro–Wilk test indicated significant deviations from normality for most variables. In contrast, Q–Q plots suggested approximate normality for all variables except 3-year-olds’ verbal ability, which showed noticeable deviation from the normal distribution. Given the robustness of Spearman correlations and regression analyses to non-normal distributions and considering the theoretical importance of the abovementioned variable, we retained it in the analysis. Accordingly, we used Spearman rank-order correlations for all bivariate analyses to ensure robustness against non-normal distributions.

Before the analyses, we determined missing values, finding two in 3-year-olds’ sleep measures, one in the mother’s educational level, one in the father’s educational level, four in annual household income, and one missing score of a 3-year-old’s ASQ-a (verbal ability). For variables other than the 3-year-old’s ASQ-a, we used multiple imputation and created five multiply imputed datasets in fully conditional specification, using predictive mean matching. We were unable to impute a missing value for the 3-year-old’s ASQ-a, so we used pairwise deletion instead as the ASQ-a score had already been calculated using multiple items and had been corrected to account for nonresponse.

### 2.3. Measures

#### 2.3.1. Sleep Questionnaire

To measure infant sleep duration at ages 1, 1.5, and 3 years, parents were asked to indicate when their infant had slept on the previous day by drawing lines through boxes indicating 30 min intervals from 12:00 a.m. to 12:00 a.m. of the next day. We totaled the sleep time in increments of 30 min between 7 p.m. and 7 a.m. as night sleep duration. If the children went to bed earlier than 7 p.m. or got up later than 7 a.m., we adjusted the thresholds in these cases to reflect the children’s actual night sleep, regarding it as night sleep unless they stayed awake for more than an hour between 7 p.m. and 7 a.m. We totaled the sleep time in increments of 30 min all day (24 h) as total sleep duration. We calculated day sleep duration by subtracting night sleep duration from total sleep duration. Sleep duration was assessed based on parental recall of a single day; this method may be less accurate than multi-day sleep diaries used in other studies. Parents were not instructed to keep a concurrent diary or to log each awakening, and the method of determining wakefulness may have varied among respondents. Partial awakenings within a 30 min interval were recorded based on the parent’s overall judgment; if the child was perceived to be mostly asleep, the interval was marked as sleep.

#### 2.3.2. Yogurt Intake Questionnaire

To assess yogurt intake frequency at age 1 year, mothers responded to the question “How many times a week does your child have yogurt?” based on their recall of the most recent week, selecting from the following options: does not eat yogurt, 1 or 2 times per week, 3 or 4 times per week, 5 or 6 times per week, 1 time per day, 2 times per day, and at least 3 times per day. Responses were scored from 0 to 6 depending on frequency (e.g., 0 points for “not eating” and 6 points for “at least 3 times per day”).

#### 2.3.3. Digit Span

At age 4 years, the children’s WM was assessed with the digit span task on the Kyoto Scale of Psychological Development 2001 (Kyoto International Social Welfare Exchange Center, Kyoto, Japan). In this task, the children repeated strings of digits in the same order used by the experimenter. This test began with a two-digit string, and the length was increased if the children succeeded in one of three trials. When the string increased to four digits, the experimenter further increased string length if the children succeeded in one of two trials. The string length was increased up to six digits. We defined the digit span score as the highest number of digits that children could successfully recall; if the children failed all two-digit trials, we assigned a score of 1. Digit span tasks have been shown to be a reliable and valid measure of WM in preschool-aged children [22], supporting the appropriateness of this assessment at age 4.

#### 2.3.4. Verbal Ability

As a measure of verbal ability at age 3, we used the ASQ-3 communication subscale, Japanese version (Igaku-Shoin Ltd., Tokyo, Japan), the reliability and validity of which (including its communication domain used to assess verbal ability) have been confirmed in a validation study reporting adequate internal consistency and test–retest reliability [23]. Specifically, the internal consistency of the communication subscale (Cronbach’s alpha) ranged from 0.62 to 0.89, and the test–retest reliability (intraclass correlation coefficient) ranged from 0.75 to 0.97, indicating sufficient reliability for developmental research. The questionnaire includes six questions answered with “yes,” “sometimes,” or “not yet.” We assigned “yes” results as 10 points, “sometimes” as 5 points, and “not yet” as 0 points. The scores ranged from 0 to 60. If one or two answers were missing, we corrected the score according to the manual. If more than three of the six questions were not answered, we treated them as missing values.

### 2.4. Use of AI Tools

Microsoft Copilot (GPT-4) was used during the revision stage of manuscript preparation to assist with language editing, summarization of reviewer comments, and formatting suggestions. All AI-assisted content was carefully reviewed and verified by the authors.

## 3. Results

Table 1 presents descriptive statistics for cognitive and sleep measures, collected longitudinally from the same cohort at ages 1, 1.5, 3, and 4 years. The 4-year-olds’ digit span scores, reflecting verbal WM, averaged 3.40 (SD = 0.92), with a range of 1 to 5, indicating moderate variability in memory capacity. The 3-year-olds’ verbal ability scores showed a wider distribution (M = 53.83, SD = 9.04), suggesting substantial individual differences in language development.

Sleep duration data reveal developmental changes over time. At age 1 year, children slept an average of 12.94 h per day, with night sleep accounting for the majority (M = 10.31 h). Day sleep showed considerable variability (SD = 1.41, range = 0–8.5 h), indicating the diverse sleep patterns prevalent at this age. As children aged, total sleep duration gradually decreased: 1.5-year-olds averaged 12.48 h and 3-year-olds averaged 11.43 h. Notably, day sleep duration declined markedly from 2.62 h at age 1 to 1.54 h at age 3, reflecting a typical developmental shift toward consolidated night sleep.

Table 2 shows the distribution of yogurt intake frequency among 1-year-olds, revealing a wide range of dietary habits. Approximately 25% of the children did not consume yogurt at all, but the largest group (28.7%) consumed yogurt 1 or 2 times a week. The sample (18.3%) reported moderate intake (3–4 times/week), and 16.5% consumed yogurt daily. Only a small proportion of children (2.4%) consumed yogurt more than once a day. These results suggest that although this cohort’s yogurt intake was common, high-frequency consumption was relatively rare. First, we computed Spearman correlations between cognitive measures (4-year-olds’ verbal WM and 3-year-olds’ verbal ability) and sleep measures collected longitudinally at ages 1, 1.5, and 3 years, along with yogurt intake frequency at age 1 (Table 3). Significant positive correlation was found between 1-year-olds’ night sleep duration and 4-year-olds’ verbal WM (*r* = 0.19, *p* < 0.05), suggesting that longer night sleep in infancy supports later memory development. Additionally, a marginally significant correlation was observed between 3-year-olds’ total sleep duration and 4-year-olds’ WM (*r* = 0.13, *p* < 0.10), indicating a potential developmental link.

The frequency of yogurt intake at age 1 also showed a significant positive correlation with 4-year-olds’ verbal WM (*r* = 0.19, *p* < 0.05), supporting the hypothesis that early dietary habits influence later cognitive outcomes. In contrast, no significant correlations were found between sleep measures and 3-year-olds’ verbal ability, although 1-year-olds’ night sleep duration showed a marginal association (*r* = 0.15, *p* < 0.10). These findings suggest that early sleep and nutrition are more closely related to WM development than to verbal ability. Although we examined Spearman correlations between 1-year-olds’ yogurt intake frequency and all sleep measures, no significant associations were found (see Appendix A). This result contradicts those of a previous study that reported a relationship between 3-year-olds’ sleep measures and 1-year-olds’ yogurt intake frequency. Finally, we computed Spearman correlations of the mother’s educational level, father’s educational level, and annual household income with all variables. We found a significant correlation between annual household income and yogurt intake frequency only in 1-year-olds (*r* = 0.18, *p* = 0.03). No other significant correlations were observed.

Correlation analysis showed that 1-year-olds’ yogurt intake frequency could directly affect 4-year-olds’ verbal WM, even if the intake was not associated with 3-year-olds’ sleep measures. However, we also found a relationship between 1-year-olds’ yogurt intake frequency and annual household income; this relationship might indirectly affect the relationship with 4-year-olds’ verbal WM. Therefore, controlling for the effects of annual household income in addition to 3-year-olds’ verbal ability, we conducted a hierarchical linear regression analysis for 4-year-olds’ verbal WM to investigate how 1-year-olds’ night sleep duration and yogurt intake frequency contributed to 4-year-olds’ verbal WM. We entered annual household income in the first step and 3-year-olds’ verbal ability in the second step. In the third and fourth steps, we entered 1-year-olds’ night sleep duration and yogurt intake frequency into the equation using a forward stepwise method. The results of this hierarchical regression analysis are summarized in Table 4. They indicate that the overall model in the fourth step was significant (*F*(4, 155) = 4.76, *p* < 0.05), explaining 11% of the variance in 4-year-olds’ verbal WM scores. Verbal ability scores at age 3 years significantly contributed to variance in verbal WM scores at age 4; however, annual household income did not. After controlling for annual household income and 3-year-olds’ verbal ability, 1-year-olds’ yogurt intake frequency significantly accounted for positive variance in verbal WM scores at age 4. Finally, 1-year-olds’ night sleep duration accounted marginally for a positive variance in verbal WM scores at age 4. Although some correlations reached statistical significance, the effect sizes were small, suggesting limited practical significance.

## 4. Discussion

This study found that night sleep duration and yogurt intake frequency at age 1 year were positively associated with WM performance at age 4, suggesting that early life factors, including sleep and diet, play a role in cognitive development.

Correlation analysis showed no association between WM performance at age 4 years and total sleep duration, although it did show an association with night sleep duration at age 1. Results showing no association between total sleep duration and WM performance were consistent with those of previous studies [9,13,15]. The association found for night sleep duration at age 1 year was similar in pattern to that identified by Bernier et al. [13] but was inconsistent with that reported by Zhang et al. [15]. This discrepancy may have been influenced by differences in sleep reporting methods among studies. We obtained reports of actual sleep status at 30 min intervals the day before administering the questionnaire, whereas Zhang et al. [15] required parents to report how long their children usually slept every night and Bernier et al. [13] obtained 3-day sleep diaries. In our data, many children had multiple awakenings at 30 min increments, shortening their night sleep duration. Thus, when examining the relationship between night sleep duration and cognitive ability, reflecting night arousal may be advantageous. Alternatively, although some have suggested that napping negatively affects WM performance [15,16], we found no association between WM performance at age 4 and all day sleep durations at ages 1, 1.5, or 3 years. Thus, the study suggests that it is unlikely that nap length is directly linked to poor WM performance. Overall, the study suggests that steady night sleep for as long as possible at age 1 can facilitate WM development at preschool age.

Notably, night sleep duration at 1 year affected WM performance at 4 years, but night sleep durations at 1.5 and 3 years did not. This pattern resembles that observed by Bernier et al. [13]; the authors found that night sleep at age 1 year affected executive functions at age 4, suggesting that the 1-year-old time point is important in relation to sleep duration. Zhang et al. [15] found no association with WM for either total or night sleep duration; however, this may be because the authors treated age as a covariate in the 19–60-month-old population. Although the precise mechanism through which longer night sleep in 1-year-olds facilitates WM development remains unclear, it seems prudent to consider this factor in future studies.

WM performance at age 4 years was also significantly correlated with yogurt intake frequency at age 1 (see Table 3); nevertheless, unlike previous studies, yogurt intake frequency at age 1 was not associated with any sleep duration measures (including at age 3). Inoue et al.’s [17] differing results may stem from their consideration of the risk of shorter sleep duration. We believe that it is reasonable to speculate that yogurt intake reduces the risk of excessively short sleep duration but not to an extent that it can directly affect sleep length. However, yogurt intake frequency at age 1 year was positively correlated with household income; the greater the household income, the higher was the yogurt intake frequency at age 1. Therefore, using hierarchical multiple regression analysis and controlling for the effects of household income, we examined whether yogurt intake frequency at age 1 independently affected WM development without influencing sleep. We found that even after controlling for household income, yogurt intake frequency at age 1 was independently associated with WM performance at age 4 (Table 4, Step 4). Therefore, we can reasonably conclude that yogurt intake directly affects the development of WM without affecting sleep.

Our data reinforce the findings of Tanaka et al. [24], who identified an association between the frequency of intake of fermented foods, including yogurt, during pregnancy and ASQ scores at age 1. This study does not clarify the mechanism by which yogurt intake influences WM development, although we note previous data showing that gut microbiota affect cognitive performance [25,26] and that yogurt intake improves gut microbiota [18]. Thus, we speculate that the impact is likely mediated through gut microbiota. Data indicate that the more frequently yogurt is consumed, the lower is the concentration of toxic substances in children’s hair [27]. The impact of yogurt, a fermented food, on gut microbiota may be considerable [18].

This study has several limitations. First, the relatively small sample size may limit the generalizability of the findings [28]. Second, sleep duration was assessed via parent-reported questionnaires, which may be subject to recall bias [29]. Third, while we controlled for household income, other potential confounding factors such as parental education or home environment were not considered [30]. Future studies with larger, more diverse samples and objective sleep measures are warranted.

## 5. Conclusions

This study examined the effects of day, night, and total sleep durations at ages 1, 1.5, and 3 years on WM at age 4. The analysis also considered the association between yogurt intake frequency at age 1 and sleep duration. Unlike previous research, we did not find an association between yogurt intake frequency and sleep duration; however, multiple regression analysis revealed that yogurt intake frequency and night sleep duration at age 1 year were each independently associated with WM performance at age 4. These findings suggest that frequent yogurt intake at age 1 and longer night sleep can lead to higher WM performance at age 4. Given this study’s relatively small sample size, future studies are needed to replicate and confirm these findings in larger, more diverse populations.

## Figures and Tables

**Table 1 nutrients-17-03081-t001:** Means, standard deviations, and ranges for all cognitive and sleep measures.

	M	SD	Range
4-year-olds’ digit span: working memory (digits)	3.40	0.92	1–5
3-year-olds’ verbal ability (points)	53.83	9.04	10–60
1-year-olds’ total sleep duration (hours)	12.94	1.67	8–18.5
1-year-olds’ night sleep duration (hours)	10.31	0.99	6–12.5
1-year-olds’ day sleep duration (hours)	2.62	1.41	0–8.5
1.5-year-olds’ total sleep duration (hours)	12.48	1.31	8.5–16
1.5-year-olds’ night sleep duration (hours)	10.28	0.90	7–13
1.5-year-olds’ day sleep duration (hours)	2.20	1.17	0–5.5
3-year-olds’ total sleep duration (hours)	11.43	1.11	8.5–14.5
3-year-olds’ night sleep duration (hours)	9.89	0.84	7–12.5
3-year-olds’ day sleep duration (hours)	1.54	1.08	0–3.5

**Table 2 nutrients-17-03081-t002:** Degree distribution of 1-year-olds’ yogurt intake frequency.

	N	N%
Not eating	41	25
1 or 2 times/week	47	28.7
3 or 4 times/week	30	18.3
5 or 6 times/week	15	9.1
1 time/day	27	16.5
2 times/day	2	1.2
At least 3 times/day	2	1.2

**Table 3 nutrients-17-03081-t003:** Spearman correlations of 4-year-olds’ working memory and 3-year-olds’ verbal ability with sleep measures and 1-year-olds’ yogurt intake frequency.

	4-Year-Olds’ Working Memory	3-Year-Olds’ Verbal ability
1-year-olds’ total sleep duration (hours)	0.07	0.11
1-year-olds’ night sleep duration (hours)	0.19 *	0.15 ^†^
1-year-olds’ day sleep duration (hours)	−0.08	0.02
1.5-year-olds’ total sleep duration (hours)	−0.01	0.10
1.5-year-olds’ night sleep duration (hours)	0.08	0.11
1.5-year-olds’ day sleep duration (hours)	−0.07	0.03
3-year-olds’ total sleep duration (hours)	0.13 ^†^	0.06
3-year-olds’ night sleep duration (hours)	0.03	0.04
3-year-olds’ day sleep duration (hours)	0.09	0.01
1-year-olds’ frequency of yogurt intake (points)	0.19 *	0.03

* *p* < 0.05. ^†^ *p* < 0.10.

**Table 4 nutrients-17-03081-t004:** Hierarchical regression results for working memory at age 4 years.

Variable	*B*	*95% CI for B*		*SE B*	*β*	*R* ^2^	∆*R*^2^
Step 1						0.00	−0.00
Constant	3.47 *	3.09	3.85	0.20			
Annual household income	−0.02	−0.12	0.08	0.05	−0.04		
Step 2						0.07	0.06 *
Constant	2.09 *	1.20	2.98	0.46			
Annual household income	−0.02	−0.12	0.07	0.05	−0.05		
Verbal ability at 3 years	0.03 *	0.01	0.04	0.00	0.26 *		
Step 3						0.09	0.08 *
Constant	1.98 *	1.09	2.89	0.45			
Annual household income	−0.04	−0.13	0.05	0.05	−0.08		
Verbal ability at 3 years	0.03 *	0.01	0.04	0.00	0.26 *		
Frequency of yogurt intake at 1 year	0.10 *	0.01	0.19	0.05	0.16 *		
Step 4						0.11	0.09 ^†^
Constant	0.82	−0.79	2.42	0.82			
Annual household income	−0.04	−0.13	0.05	0.05	−0.08		
Verbal ability at 3 years	0.03 *	0.01	0.04	0.00	0.25 *		
Frequency of yogurt intake at 1 year	0.10 *	0.01	0.19	0.05	0.16 *		
Night sleep duration at 1 year	0.12 ^†^	−0.02	0.26	0.07	0.13 ^†^		

CI = confidence interval; * *p* < 0.05. ^†^ *p* < 0.10.

## Data Availability

The data used in this study are unsuitable for public deposition due to ethical restrictions and the legal framework of Japan. Their publication is prohibited by the Act on the Protection of Personal Information (Act No. 57 of 30 May 2003, amendment on 9 September 2015) to publicly deposit the data containing personal information. The Ethical Guidelines for Medical and Health Research Involving Human Subjects enforced by the Japan Ministry of Education, Culture, Sports, Science and Technology and the Ministry of Health, Labor and Welfare also restrict the open sharing of epidemiological data. All inquiries about access to data should be sent to jecs-en@nies.go.jp. The person responsible for handling enquiries sent to this e-mail address is Dr. Shoji F. Nakayama, JECS Program Office, National Institute for Environmental Studies.

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
