# Peer review of "Association Between Working Memory at Age 4 Years and Night Sleep Duration and Yogurt Intake Frequency at Age 1 Year"

_nutrients, 2025, doi:10.3390/nu17193081_

Round 1

Reviewer 1 Report

Comments and Suggestions for Authors

General: This prospective observational study examined the relationship between working memory (WM) at 4 years of age with sleep duration at 1,1.5,and 3 years of age and with yoghurt intake at 1 year of age using Spearman rank correlation analysis and then hierarchical linear regression including other variables found significant in Spearman analysis.  The findings support those of some prior studies while adding an additional dimension.  However, generalizability is unclear given select nature of the population unstated limitations, and clinical significance is unclear.  Revisions are necessary before this can be considered for publication.

Abstract: the use of the word “explained” to describe associations is too strong; “associated with” would be better; even better “associated with to some (or limited degree).

Introduction:  This section provides some relevant background but does not adequately identify the gap this study is designed to address (only partially in lines65-66).  Lines 35-36 should cite evidence supporting this assertion rather than stating it is “reasonable to assert.”  In lines 51-55, the conversely statement appears to partially contradict the prior sentence. 

Materials and Methods:  In line 107 the measure of sleep duration is reportedly based on one day’s parental recall.  The accuracy or limitations of such recall need to be addressed and justified compared with recall of multiple days used in one other study.  There is no indication parents were instructed to keep a concurrent diary of each time child awakened, nor was there any stipulation as to how much variance there may have been in parents method of detection of waking.  It is also unclear how partial awakening during a given 30 minute interval was rated: asleep vs. awake.  In Section 2.3.2, yoghurt intake is based on recall of a week’s intake, but it is not clear whether this is the most recent week or an average of several weeks.  In section 2.3.3 re digit span, no information is provided regarding accuracy/reliability of this measure of working memory at 4 years of age. The word “increased” does not match the tense of “begins,” which should probably be “began” and and “increased” should be passive (“was increased”) or rewording using active voice.  In 2.3.4, ASQ-3 is used as a measure of verbal ability at age 3, but there is no explanation as to the accuracy/reliability of this screening tool in assessing verbal ability.  There is no section describing statistical methods or rationale for the methods used; rather this is included to some extent in Results.

Results: The first paragraph (lines 143-152) in part reports methodology for dealing with missing values, which would better be included in Methods section.  In line 162, Spearman correlations were reported conducted for 3 sociological/demographic variables, but it is not clear whether these were only done in relation to yoghurt intake or whether also done in relation to sleep times at various ages, since no table of such results is included.  Table 1 would be more readable if all variables in the first column were left justified rather than centered; likewise for Tables 2 and 3. Though three correlations shown in Table 3 may be statistically significant (only one p<0.05), the magnitude of the correlations are rather low.  Likewise for the correlations shown in Table 4 for the hierarchical regression analysis.

Discussion/Conclusion:.   In line 205, “and was inconsistent” should be “but was inconsistent”.  Limitations of this study need to be described.  Seem comment on abstract above regarding use of the word “explained”.  The clinical significance of the findings of this study are unclear.

Comments on the Quality of English Language

Suggested revisions of a few key words

Author Response

Comments 1: General: This prospective observational study examined the relationship between working memory (WM) at 4 years of age with sleep duration at 1,1.5,and 3 years of age and with yoghurt intake at 1 year of age using Spearman rank correlation analysis and then hierarchical linear regression including other variables found significant in Spearman analysis.  The findings support those of some prior studies while adding an additional dimension.  However, generalizability is unclear given select nature of the population unstated limitations, and clinical significance is unclear.  Revisions are necessary before this can be considered for publication.

Response 1: Thank you for your comprehensive feedback. We agree with your assessment and have made several revisions to address these concerns.

First, we have added a paragraph explicitly discussing the limitations of the study, including the relatively small sample size, the use of parent-reported sleep data (which may be subject to recall bias), and the omission of certain potential confounding variables such as parental education and home environment. This addition has been made on page 8, lines 309–314.

Second, we have revised the Conclusion section to better emphasize the contribution of the study to current understanding, particularly regarding the independent associations of yogurt intake and night sleep duration at age 1 with WM performance at age 4. Finally, given the limited sample size, we have highlighted the need for future studies to replicate and confirm these findings. These revisions appear on page 9, lines 321–324.

“This study has several limitations. First, the relatively small sample size may limit the generalizability of the findings. Second, sleep duration was assessed via parent-reported questionnaires, which may be subject to recall bias. Third, while we controlled for household income, other potential confounding factors such as parental education or home environment were not considered. Future studies with larger, more diverse samples and objective sleep measures are warranted.”

“These findings suggest that frequent yogurt intake at age 1 and longer night sleep can lead to higher WM performance at age 4. Given this study’s relatively small sample size, future studies can replicate and confirm these findings in larger, more diverse populations.”

Comments 2: Abstract: the use of the word “explained” to describe associations is too strong; “associated with” would be better; even better “associated with to some (or limited degree).

Response 2: Thank you for pointing this out. We agree with your observation. Therefore, we have revised the sentence in the abstract to avoid implying causality and to better reflect the nature of the observed relationship. Specifically, we have replaced the phrase “explained WM performance” with “was associated with WM performance to a limited degree.”

This change can be found on page 1, line 25–27 of the revised manuscript.

“To a limited degree, both yogurt intake frequency and night sleep duration at 1 year were associated with WM performance at 4 years,”

Comments 3: Introduction:  This section provides some relevant background but does not adequately identify the gap this study is designed to address (only partially in lines65-66).  Lines 35-36 should cite evidence supporting this assertion rather than stating it is “reasonable to assert.”  In lines 51-55, the conversely statement appears to partially contradict the prior sentence.

Response 3: Thank you for highlighting this; we agree with your comment. Accordingly, to clarify the research gap, we have explicitly mentioned the lack of studies examining the interaction between yogurt intake and sleep duration on WM performance in children (page 2, lines 66–70); furthermore, we have made certain revisions to include supporting citations (page 2, lines 35–37) and modifications to avoid contradiction and ensure clarity (page 2, lines 52–54).

“These functions are considered a prerequisite for higher intellectual activity and serve as determinants of the extent to which adults can eventually perform complex cognitive tasks [3,4].”

“However, other studies have reported that neither total nor night sleep duration significantly affects WM performance and that children who do not nap may perform better on WM tasks [15].”

“Despite increasing academic interest in the gut–brain axis and its potential influence on cognitive development, no previous study has investigated how yogurt intake—a dietary factor known to affect gut microbiota—interacts with sleep duration to influence children’s WM performance. This study aims to address this critical gap in the literature.”

Comments 4: Materials and Methods: In line 107 the measure of sleep duration is reportedly based on one day’s parental recall.  The accuracy or limitations of such recall need to be addressed and justified compared with recall of multiple days used in one other study. There is no indication parents were instructed to keep a concurrent diary of each time child awakened, nor was there any stipulation as to how much variance there may have been in parents method of detection of waking.  It is also unclear how partial awakening during a given 30 minute interval was rated: asleep vs. awake.

Response 4: Thank you for your suggestion. To address it, we have now described the limitations of using one-day parental recall for measuring sleep duration and explained how partial awakenings were treated. Specifically, we have added the following sentences to the revised manuscript on page 4, lines 145–150:

“Sleep duration was assessed based on parental recall of a single day; this method may be less accurate than multi-day sleep diaries used in other studies. Parents were not instructed to keep a concurrent diary or to log each awakening, and the method of determining wakefulness may have varied among respondents. Partial awakenings within a 30-minute interval were recorded based on the parent’s overall judgment; if the child was perceived to be mostly asleep, the interval was marked as sleep.”

Comments 5: In Section 2.3.2, yoghurt intake is based on recall of a week’s intake, but it is not clear whether this is the most recent week or an average of several weeks.

Response 5: Thank you for your feedback. We agree that the original description of the yogurt intake assessment lacked clarity regarding the recall period. To address this, we have revised the relevant sentence in Section 2.3.2 to specify that parents were instructed to recall intake in the most recent week.

This change can be found on page 4, lines 152–154 of the revised manuscript.

“To assess yogurt intake frequency at age 1 year, mothers responded to the question “How many times a week does your child have yogurt?” based on their recall of the most recent week, selecting from the following options:”

Comments 6: In section 2.3.3 re digit span, no information is provided regarding accuracy/reliability of this measure of working memory at 4 years of age. The word “increased” does not match the tense of “begins,” which should probably be “began” and and “increased” should be passive (“was increased”) or rewording using active voice.

Response 6: Thank you for highlighting this issue. We have made changes to ensure tense consistency and improve clarity—we have revised “begins” to “began” and rephrased the text for the appropriate voice. The changes can be found on page 4, section 2.3.3, lines 161–162. Additionally, we have added a statement regarding the reliability and validity of the digit span task for assessing working memory in preschool-aged children, supported by a newly added reference [22], on page 4, lines 166–168.

“This test began with a two-digit string, and the length was increased if the children succeeded in one of three trials.”

“Digit span tasks have been shown to be a reliable and valid measure of WM in preschool-aged children [22], supporting the appropriateness of this assessment at age 4.”

Comments 7: In 2.3.4, ASQ-3 is used as a measure of verbal ability at age 3, but there is no explanation as to the accuracy/reliability of this screening tool in assessing verbal ability.  There is no section describing statistical methods or rationale for the methods used; rather this is included to some extent in Results.

Response 7: Thank you for your feedback. We have revised the manuscript to clarify the statistical methods and their rationale.

Additionally, we have expanded the Procedure section (page 3, lines 112–133) to include a detailed description of the statistical procedures used in the study, including power analysis, normality testing, missing data handling, and the rationale for using Spearman correlations. However, we would like to clarify that reliability analyses of ASQ-3 scores (e.g., Cronbach’s alpha or ICC) were not conducted in this study.

To address concerns regarding the validity of the ASQ-3 as a measure of verbal ability, we have added the following sentence to Section 2.3.4 Verbal Ability (page 4, line 170-173):

“As a measure of verbal ability at age 3, we used the ASQ-3 communication subscale, Japanese version, the reliability and validity of which (including its communication domain used to assess verbal ability) have been confirmed in a validation study reporting adequate internal consistency and test–retest reliability [23].”

Comments 8: Results: The first paragraph (lines 143-152) in part reports methodology for dealing with missing values, which would better be included in Methods section.

Response 8: Thank you for this suggestion; we agree with it. We have moved the description regarding the handling of missing values to the end of the Procedure subsection in the Methods section (page 3, lines 126–133).

Comments 9: In line 162, Spearman correlations were reported conducted for 3 sociological/demographic variables, but it is not clear whether these were only done in relation to yoghurt intake or whether also done in relation to sleep times at various ages, since no table of such results is included.

Response 9: Thank you for pointing this out. We have clarified the scope of the Spearman correlation analyses and explained why no table was included. As no significant correlations were found between sleep measures and the sociological/demographic variables, a table was not created. Specifically, we have added the following sentences to the revised manuscript on page 5, lines 220–221 and 226–227:

“Although we examined Spearman correlations between 1-year-olds’ yogurt intake frequency and all sleep measures, no significant associations were found.”

“No other significant correlations were observed.”

Comments 10: Table 1 would be more readable if all variables in the first column were left justified rather than centered; likewise for Tables 2 and 3.

Response 10: Thank you for your helpful suggestion. We have adjusted the alignment of the variable names in the first columns of Tables 1, 2, and 3 to be left justified instead of center-aligned to improve readability. These changes can be found in the revised manuscript on pages 5–6.

Comments 11: Though three correlations shown in Table 3 may be statistically significant (only one p<0.05), the magnitude of the correlations are rather low.  Likewise for the correlations shown in Table 4 for the hierarchical regression analysis.

Response 11: Thank you for your insightful comment. We agree with your observation. To address this, we have added a note in the Results section to acknowledge that although some correlations in Table 3 and Table 4 were statistically significant, the magnitudes of these correlations were relatively low, indicating limited practical significance. This clarification has been added to page 7, lines 251–253.

“Although some correlations reached statistical significance, the effect sizes were small, suggesting limited practical significance.”

Comments 12: Discussion/Conclusion: In line 205, “and was inconsistent” should be “but was inconsistent”.  Limitations of this study need to be described.  Seem comment on abstract above regarding use of the word “explained”.  The clinical significance of the findings of this study are unclear.

Response 12: Thank you for your valuable feedback. We have made the following revisions in response.

First, we have corrected “and was inconsistent” to “but was inconsistent” on page 7, line 265.

Second, we have added a paragraph discussing the limitations of the study, including potential confounding factors, generalizability, and the observational nature of the data. This addition can be found on page 8, lines 311–316.

Third, we have revised the use of the word “explained” in the Discussion section to ensure more accurate and appropriate language, in line with the earlier comment on the abstract. The revised wording appears on page 8, line 298-299.

Finally, we have also added a sentence addressing the clinical significance of the findings, emphasizing the potential implications for early childhood dietary and sleep interventions. This can be found on page 9, lines 325–326.

 “...but was inconsistent”

“This study has several limitations. First, the relatively small sample size may limit the generalizability of the findings. Second, sleep duration was assessed via parent-reported questionnaires, which may be subject to recall bias. Third, while we controlled for household income, other potential confounding factors such as parental education or home environment were not considered. Future studies with larger, more diverse samples and objective sleep measures are warranted.”

“was independently associated with”

“Given this study’s relatively small sample size, future studies can replicate and confirm these findings in larger, more diverse populations.”

4. Response to Comments on the Quality of English Language

Point 1: Suggested revisions of a few key words.

Response 1: Thank you for your comment regarding the quality of the language. We appreciate the suggestion and have carefully revised the manuscript to improve clarity and word choice, including the suggested revisions of key words. These changes have been applied throughout the manuscript and are marked in red.

Reviewer 2 Report

Comments and Suggestions for Authors

The manuscript entitled “nutrients-3870647_ Association Between Working Memory at Age 4 Years and Night Sleep Duration and Frequency of Yogurt Intake at Age 1 Year” has been submitted for consideration in the special issue “ Nutrition and Brain Health Across the Lifespan: Insights into Mental, Cognitive, and Sleep Outcomes in Health and Disease” of the journal “Nutrients”.

Summary of the study
This study examined the relationship between sleep duration (nighttime, total, and daytime) at 1, 1.5, and 3 years of age and working memory (WM) at 4 years, measured by the forward digit span task. Given previous findings that yogurt intake at age 1 is associated with sleep duration, this variable was also included. The results indicated that both nighttime sleep duration and yogurt intake at age 1 were significantly associated with WM performance at age 4, although yogurt intake was not correlated with sleep duration in this sample. Multiple regression analyses showed that frequent yogurt intake and longer nighttime sleep at age 1 independently predicted better WM at age 4.

Evaluation

  • Title: The title is informative and reflects the content of the study.
  • Abstract: The abstract presents the study but should be expanded. It does not specify the design, the study population, or statistical results, which are necessary to strengthen its clarity and rigor.
  • Introduction: The introduction outlines the role of sleep in children’s WM. However, the section discussing yogurt intake and the microbiota (lines 65–67) should be further developed, since this is central to the study. The introduction should end with a clearly stated hypothesis and precise objectives.
  • Materials and Methods: The study is based on a cohort of 164 mother–child pairs followed up to age 4 with four assessment points (1 year, 1.5 years, 3 years, and 4 years). The questionnaires used to assess sleep quality, yogurt consumption, and children’s WM (Digit Span) must be described with bibliographic references; if developed for this study, their validation should be clarified. The methodology applied should be reported in detail, including how normality of variable distributions was assessed and whether sample size calculations were performed.
  • Results: The description of statistical software and methods should be moved to the Methods section. Units should be added to Tables 1 and 3. Tables must be adequately described in the text, highlighting the most relevant results.
  • Discussion: The discussion should begin with an interpretation of the main findings, aligned with the stated objectives, rather than restating the study aim. The comparison with existing literature is well addressed, but the strengths and limitations of the study should be explicitly discussed, including the relatively small sample size.
  • Conclusion: Conclusions should emphasize the contribution of the study to current knowledge. It would also be appropriate to highlight the need for future studies to replicate and confirm these findings, given the robust design but relatively limited sample size.

Author Response

Comments 1: Title: The title is informative and reflects the content of the study.

Response 1: We appreciate your positive feedback on the title.

Comments 2: Abstract: The abstract presents the study but should be expanded. It does not specify the design, the study population, or statistical results, which are necessary to strengthen its clarity and rigor.

Response 2: Thank you for your valuable comment. We agree with your suggestion. Therefore, we have revised the abstract to include additional details regarding the study design, study population, and key statistical results to improve clarity and rigor. The updated abstract can be found on page 1, lines 16–29.

Background/Objectives: This cohort study examined the effects of sleep durations (night, day, and total daily) at ages 1, 1.5, and 3 years on working memory (WM) assessed at age 4, measured using forward digit span. Methods: Because frequency of yogurt intake at 1 year has been shown to affect sleep duration at 3 years, we also accounted for the association between frequency of yogurt intake at 1 year and sleep duration, based on a recent study indicating positive effects of yogurt on sleep. The study included 164 mother–child pairs observed from ages 1 to 4. Results: Spearman correlations and hierarchical multiple regression analyses revealed that both night sleep duration and yogurt intake at age 1 were significantly associated with WM performance at age 4. In this sample, however, yogurt intake was not correlated with sleep duration. Conclusions: To a limited degree, both yogurt intake frequency and night sleep duration at 1 year were associated with WM performance at 4 years, indicating that frequent yogurt intake at 1 year and longer night sleep duration may lead to higher WM performance at 4 years.”

Comments 3: Introduction: The introduction outlines the role of sleep in children’s WM. However, the section discussing yogurt intake and the microbiota (lines 65–67) should be further developed, since this is central to the study. The introduction should end with a clearly stated hypothesis and precise objectives.

Response 3: Thank you for your suggestion. In response, we have expanded the section on yogurt intake and the microbiota by explaining the role of the gut–brain axis and how yogurt, as a source of probiotics, may influence sleep and cognitive functions, including WM. This revision can be found on page 2, lines 62–66, and includes additional references [17,18]. Furthermore, we have clarified the research gap by explicitly stating the lack of studies examining the interaction between yogurt intake and sleep duration on WM performance in children (page 2, lines 66–70). Finally, we have added a clearly stated hypothesis and specific objectives at the end of the introduction (page 2, lines 71–73).

“This relationship may be explained by the gut–brain axis, a bidirectional communication system between the gastrointestinal tract and the central nervous system. Yogurt, as a source of probiotics, can positively influence gut microbiota composition, which in turn potentially affects sleep patterns and cognitive functions, including WM [18–20].”

“Despite increasing academic interest in the gut–brain axis and its potential influence on cognitive development, no previous study has investigated how yogurt intake—a dietary factor known to affect gut microbiota—interacts with sleep duration to influence children’s WM performance. This study aims to address this critical gap in the literature.”

“Based on previous findings, we hypothesize that yogurt intake at age 1 year positively influences sleep duration at later ages and that a longer sleep duration is associated with better WM performance at age 4 years. The study aimed to examine these relationships using longitudinal data and behavioral assessments.”

Comments 4: Materials and Methods: The study is based on a cohort of 164 mother–child pairs followed up to age 4 with four assessment points (1 year, 1.5 years, 3 years, and 4 years). The questionnaires used to assess sleep quality, yogurt consumption, and children’s WM (Digit Span) must be described with bibliographic references; if developed for this study, their validation should be clarified. The methodology applied should be reported in detail, including how normality of variable distributions was assessed and whether sample size calculations were performed.

Response 4: Thank you for your helpful comment regarding the description of the questionnaires and methodological details.

In response, we have revised the Materials and Methods section to clarify that all measures used to assess sleep quality, yogurt consumption, and children’s working memory (Digit Span) were derived from the Japan Environment and Children’s Study (JECS). These measures have been widely used in large-scale cohort studies such as JECS, and we have cited the relevant source [17] to support this.

Additionally, we have added a justification for the sample size. Specifically, we conducted an a priori power analysis using G*Power software (version 3.1), which confirmed that our sample size of 164 participants is sufficient to detect a medium effect size (ρ = 0.3) with 80% power at an alpha level of 0.05.

Furthermore, we have described the procedures used to assess the normality of variable distributions, including both the Shapiro–Wilk test and visual inspection via Q–Q plots. Based on these assessments, we used Spearman rank–order correlations for all bivariate analyses to ensure robustness against non-normal distributions.

This revision can be found on page 3, lines 110–125.

“All measures were derived from the JECS and have been widely used in similar large-scale cohort studies [21].

To ensure adequate statistical power, we conducted a priori power analysis using G*Power software (version 3.1). The analysis indicated that a sample size of 164 participants is sufficient to detect a medium effect size (ρ = 0.3) in correlation analyses with 80% power at an alpha level of 0.05. This confirms that our study is appropriately powered to detect meaningful associations among the variables of interest. For all analyses, we used SPSS version 29 (IBM Corp., Armonk, NY). Prior to handling missing data, we assessed all variables’ normality using both the Shapiro–Wilk test and visual inspection via Q–Q plots. The Shapiro–Wilk test indicated significant deviations from normality for most variables. In contrast, Q–Q plots suggested approximate normality for all variables except 3-year-olds’ verbal ability, which showed noticeable deviation from the normal distribution. Given the robustness of Spearman correlations and regression analyses to non-normal distributions and considering the theoretical importance of the abovementioned variable, we retained it in the analysis. Accordingly, we used Spearman rank–order correlations for all bivariate analyses to ensure robustness against non-normal distributions.”

Comments 5: Results: The description of statistical software and methods should be moved to the Methods section.

Response 5: Thank you for your suggestion. We have moved the description of the statistical software and analytical methods from the Results section to the end of the Procedure subsection in the Methods section to improve the logical structure of the manuscript. This revision can be found on page 3, lines 112–133.

Comments 6: Units should be added to Tables 1 and 3. Tables must be adequately described in the text, highlighting the most relevant results.

Response 6: Thank you for this feedback. We have now added units to Tables 1 and 3 and revised the main text to provide more detailed descriptions of the tables, highlighting the most relevant results.

Units added:

In Table 1, units such as “(digits),” “(points),” and “(hours)” have been added to clarify the measurement scales for cognitive and sleep variables.

In Table 3, sleep measures are labeled with “(hours)” and yogurt intake is labeled with “(points)” to indicate the scale used for frequency. The correlation coefficients are clarified as Spearman’s r values.

Textual revisions:

A detailed description of Table 1 has been added on page 4, lines 185–198, explaining the developmental changes in sleep duration and variability in cognitive scores.

A summary of Table 2 has been added on page 5, lines 199–205, describing the distribution of yogurt intake frequency.

A description of Table 3 has been added on page 5, lines 205–222, highlighting significant and marginal correlations between early sleep and dietary measures with later cognitive outcomes.

A reference to Table 4 has been added on page 6, lines 242–243, to clarify that the hierarchical regression results are presented there.

“Table 1 presents descriptive statistics for cognitive and sleep measures, collected longitudinally from the same cohort at ages 1, 1.5, 3, and 4 years. The 4-year-olds’ digit span scores, reflecting verbal WM, averaged 3.40 (SD = 0.92), with a range of 1 to 5, indicating moderate variability in memory capacity. The 3-year-olds’ verbal ability scores showed a wider distribution (M = 53.83, SD = 9.04), suggesting substantial individual differences in language development.

Sleep duration data reveal developmental changes over time. At age 1 year, children slept an average of 12.94 hours per day, with night sleep accounting for the majority (M = 10.31 hours). Day sleep showed considerable variability (SD = 1.41, range = 0–8.5 hours), indicating the diverse sleep patterns prevalent at this age. As children aged, total sleep duration gradually decreased: 1.5-year-olds averaged 12.48 hours and 3-year-olds averaged 11.43 hours. Notably, day sleep duration declined markedly from 2.62 hours at age 1 to 1.54 hours at age 3, reflecting a typical developmental shift toward consolidated night sleep.”

“Table 2 shows the distribution of yogurt intake frequency among 1-year-olds, revealing a wide range of dietary habits. Approximately 25% of the children did not consume yogurt at all, but the largest group (28.7%) consumed yogurt 1 or 2 times a week. The sample (18.3%) reported moderate intake (3–4 times/week), and 16.5% consumed yogurt daily. Only a small proportion of children (2.4%) consumed yogurt more than once a day. These results suggest that although this cohort’s yogurt intake was common, high-frequency consumption was relatively rare.”

“First, we computed Spearman correlations between cognitive measures (4-year-olds’ verbal WM and 3-year-olds’ verbal ability) and sleep measures collected longitudinally at ages 1, 1.5, and 3 years, along with yogurt intake frequency at age 1 (Table 3). Significant positive correlation was found between 1-year-olds’ night sleep duration and 4-year-olds’ verbal WM (r = .19, p < .05), suggesting that longer night sleep in infancy supports later memory development. Additionally, a marginally significant correlation was observed between 3-year-olds’ total sleep duration and 4-year-olds’ WM (r = .13, p < .10), indicating a potential developmental link.

The frequency of yogurt intake at age 1 also showed a significant positive correlation with 4-year-olds’ verbal WM (r = .19, p < .05), supporting the hypothesis that early dietary habits influence later cognitive outcomes. In contrast, no significant correlations were found between sleep measures and 3-year-olds’ verbal ability, although 1-year-olds’ night sleep duration showed a marginal association (r = .15, p < .10). These findings suggest that early sleep and nutrition are more closely related to WM development than to verbal ability.”

“The results of this hierarchical regression analysis are summarized in Table 4.”

Comments 7: Discussion: The discussion should begin with an interpretation of the main findings, aligned with the stated objectives, rather than restating the study aim. The comparison with existing literature is well addressed, but the strengths and limitations of the study should be explicitly discussed, including the relatively small sample size.

Response 7: Thank you for your thoughtful comments. We agree with your suggestions. Accordingly, we have made the following revisions.

First, we have revised the beginning of the Discussion section; this section, which earlier restated study objectives, now starts with an interpretation of the main findings in relation to the objectives. This change can be found on page 7, lines 256–258.

Second, we have added a paragraph explicitly discussing the strengths and limitations of the study, including the relatively small sample size, the use of parent-reported sleep data, and the lack of certain covariates such as parental education and home environment. These revisions appear on page 8, lines 310–315.

“This study found that night sleep duration and yogurt intake frequency at age 1 year were positively associated with WM performance at age 4, suggesting that early life factors, including sleep and diet, play a role in cognitive development.”

“This study has several limitations. First, the relatively small sample size may limit the generalizability of the findings. Second, sleep duration was assessed via parent-reported questionnaires, which may be subject to recall bias. Third, while we controlled for household income, other potential confounding factors such as parental education or home environment were not considered. Future studies with larger, more diverse samples and objective sleep measures are warranted.”

Comments 8: Conclusion: Conclusions should emphasize the contribution of the study to current knowledge. It would also be appropriate to highlight the need for future studies to replicate and confirm these findings, given the robust design but relatively limited sample size.

Response 8: Thank you for your helpful comment. We agree with your suggestions. We have revised the Conclusion section to emphasize the contribution of the study to current knowledge, particularly regarding the potential influence of early-life sleep and dietary factors on later cognitive development. Additionally, we have added a statement highlighting the need for future studies to replicate and confirm these findings, considering the relatively small sample size. These revisions can be found on page 8, lines 317–325.

“This study examined the effects of day, night, and total sleep durations at ages 1, 1.5, and 3 years on WM at age 4. The analysis also considered the association between yogurt intake frequency at age 1 and sleep duration. Unlike previous research, we did not find an association between yogurt intake frequency and sleep duration; however, multiple regression analysis revealed that yogurt intake frequency and night sleep duration at age 1 year were each independently associated with WM performance at age 4. These findings suggest that frequent yogurt intake at age 1 and longer night sleep can lead to higher WM performance at age 4. Given this study’s relatively small sample size, future studies can replicate and confirm these findings in larger, more diverse populations.”

Round 2

Reviewer 1 Report

Comments and Suggestions for Authors

Authors have adequately addressed issues/recommendations raised by this reviewer.  Only two minor changes are needed.  In the revised abstract, the following clause is out of place and should be deleted: These functions are considered a prerequisite for higher intellectual activity

Secondly, the last sentence of the Conclusion should change the word "can" to "are needed to"

Author Response

Comments 1: Authors have adequately addressed issues/recommendations raised by this reviewer.  Only two minor changes are needed.  In the revised abstract, the following clause is out of place and should be deleted: These functions are considered a prerequisite for higher intellectual activity.

Response 1: Thank you very much for your positive evaluation and for your helpful comments.

As per your suggestion, we have deleted the clause “These functions are considered a prerequisite for higher intellectual activity” from the revised abstract.

We appreciate your careful review and believe that these changes have improved the clarity of our manuscript.

Comments 2: Secondly, the last sentence of the Conclusion should change the word "can" to "are needed to".

Response 2: Thank you for your valuable suggestion.

In accordance with your comment, we have revised the last sentence of the Conclusion by changing the word “can” to “are needed to.”

We appreciate your careful review and believe this change has improved the accuracy of our manuscript (page 9, lines 329–330).

Reviewer 2 Report

Comments and Suggestions for Authors

The authors have done an excellent job in clarifying and improving the presentation of the study, which contributes to a better understanding of its content. However, they have not followed the recommendation that the introduction should conclude with a clear statement of the study objective. I suggest that they implement this change.

In the Materials and Methods section, it is still unclear which population was initially enrolled and how many participants were lost to follow-up before reaching the final sample of 165 mother–child pairs. It would also be advisable to specify the inclusion criteria for this cohort in order to assess the representativeness of the study sample.

Overall, the study is now more robust and presents its findings more clearly. Specifically, it analyzed the relationship between sleep duration (daytime, nighttime, and total) in early childhood and working memory (WM) at 4 years of age. No association was found between yogurt intake and sleep; however, both the frequency of yogurt consumption and nighttime sleep duration at 1 year were independently associated with better WM performance at age 4.

Author Response

Comments 1: The authors have done an excellent job in clarifying and improving the presentation of the study, which contributes to a better understanding of its content. However, they have not followed the recommendation that the introduction should conclude with a clear statement of the study objective. I suggest that they implement this change.

Response 1: Thank you for pointing this out. We agree with this comment. Although the Introduction included a description of the study’s hypotheses and analytical approach, we understand the importance of concluding with a clear and concise statement of the study objectives.

Accordingly, we have added the following sentence to the end of the Introduction (page 2, lines 82–84):

“Therefore, this study aimed to investigate the longitudinal associations between sleep duration at ages 1, 1.5, and 3 and working memory performance at age 4, as well as the association between yogurt intake frequency and sleep duration.”

Comments 2: In the Materials and Methods section, it is still unclear which population was initially enrolled and how many participants were lost to follow-up before reaching the final sample of 165 mother–child pairs. It would also be advisable to specify the inclusion criteria for this cohort in order to assess the representativeness of the study sample.

Response 2: Thank you for pointing this out. As you correctly noted, as this was an adjunct study, we were only granted limited access to the data of the 169 participants in the sub-cohort study at age 4, due to personal information protection policies. Therefore, we are unable to report how many participants were lost to follow-up before reaching the final sample of 165 mother–child pairs. We only excluded the data of four participants who did not complete the digit-span task.

In response to your comment, we have added a clarification to the Materials and Methods section to explain the initial number of participants and the exclusion process.

We have added the following sentence in page 2, lines 89–91:

“The sample included 165 mother–child dyads (81 boys, 84 girls) who participated in the Sub-Cohort Study of the Kyoto Regional Centre of the Japan Environment and Children’s Study (JECS) [21]; the Sub-Cohort Study initially included 169 participants at age 4 years. However, the present study excluded 4 children who did not complete the digit span task of the Kyoto Scale of Psychological Development 2001.”

Comments 3: Overall, the study is now more robust and presents its findings more clearly. Specifically, it analyzed the relationship between sleep duration (daytime, nighttime, and total) in early childhood and working memory (WM) at 4 years of age. No association was found between yogurt intake and sleep; however, both the frequency of yogurt consumption and nighttime sleep duration at 1 year were independently associated with better WM performance at age 4.

Response 3: Thank you for your positive evaluation of our revised manuscript. We appreciate your recognition of the improved clarity and robustness of the study and agree with your summary of the findings.

As noted, this study aimed to analyze the relationship between sleep duration (daytime, nighttime, and total) in early childhood and working memory at age 4. While no association was found between yogurt intake and sleep, we found that the frequency of yogurt consumption and nighttime sleep duration at age 1 were independently associated with better WM performance at age 4.

No textual changes were necessary in response to this comment, as the current version of the manuscript already reflects these findings clearly.